# Epidemiology, clinical characteristics, household transmission, and lethality of severe acute respiratory syndrome coronavirus-2 infection among healthcare workers in Ontario, Canada

**Kevin L. Schwartz**[1,2,3]*, **Camille Achonu**[1], **Sarah A. Buchan**[1,2], **Kevin A. Brown**[1,2], **Brenda Lee**[1], **Michael Whelan**[1], **Julie HC Wu**[1], **Gary Garber**[1,4]

**1** Public Health Ontario, Toronto, Ontario, Canada, **2** Dalla Lana School of Public Health, University of Toronto, Toronto, Ontario, Canada, **3** Unity Health Toronto, Toronto, Ontario, Canada, **4** Ottawa Hospital Research Institute, Ottawa, Ontario, Canada

* Kevin.schwartz@oahpp.ca

**Data Availability Statement:** Public Health Ontario (PHO) cannot disclose the underlying data. Doing

## Abstract

### Introduction

Protecting healthcare workers (HCWs) from Severe Acute Respiratory Syndrome Coronavirus-2 (SARS-CoV-2) is a priority to maintain a safe and functioning healthcare system. Our objective was to describe and compare the epidemiology, clinical characteristics, and lethality of SARS-CoV-2 infections among HCWs compared to non-HCWs.

### Methods

Using reportable disease data at Public Health Ontario, we conducted a population-based cross-sectional study comparing demographic, exposure, and clinical variables between HCWs and non-HCWs with SARS-CoV-2 infections as of 30 September 2020. We calculated rates of infections over time and determined the frequency of within household transmissions using natural language processing based on residential address. We evaluated the risk of death using a multivariable logistic regression model adjusting for age, sex, comorbidities, symptoms, and long-term care home exposure.

### Results

There were 7,050 (12.5%) HCW SARS-CoV-2 infections in Ontario, Canada, of whom 24.9% were nurses, 2.3% were physicians, and the remaining 72.8% other specialties, including personal support workers. Overall HCWs had an infection rate of 1,276 per 100,000 compared to non-HCWs of 346 per 100,000 (3.7 times higher). This difference decreased from a 7 times higher rate in April to no difference in September 2020. Twenty-six percent of HCWs had a household member with SARS-CoV-2 infection; 6.8% were probable acquisitions, 12.3% secondary transmissions, and 6.9% unknown direction of

so would compromise individual privacy contrary to PHO's ethical and legal obligations. Restricted access to the data may be available under conditions prescribed by the Ontario Personal Health Information Protection Act, 2004, the Ontario Freedom of Information and Protection of Privacy Act, the Tri-Council Policy Statement: Ethical Conduct for Research Involving Humans (TCPS 2 (2018)), and PHO privacy and ethics policies. Data are available for researchers who meet PHO's criteria for access to confidential data. Information about PHO's data access request process is available on-line at https://www. publichealthontario.ca/en/data-and-analysis/using-data/data-requests.

**Funding:** The author(s) received no specific funding for this work.

**Competing interests:** The authors have declared that no competing interests exist.

transmission. Death among HCWs was 0.2% compared to 6.1% of non-HCWs. The risk of death in HCWs remained significantly lower than non-HCWs after adjustment (adjusted odds ratio 0.09; 95%CI 0.05–0.17).

## Conclusion

HCWs represent a disproportionate number of diagnosed SARS-CoV-2 infections in Ontario, however this discrepancy is at least partially explained by limitations in testing earlier in the pandemic for non-HCWs. We observed a low risk of death in HCWs which could not be completely explained by other factors.

## Introduction

We are in the midst of a global pandemic from Coronavirus disease of 2019 (COVID-19), caused by the virus Severe Acute Respiratory Syndrome Coronavirus-2 (SARS-CoV-2). COVID-19 is impacting healthcare systems globally which are coping with outbreaks in congregate living facilities and the rapid influx of critically ill patients requiring care in intensive care units [1]. Preventing healthcare worker (HCW) infections is critical to maintaining a functioning healthcare system, and they have been a priority group for testing throughout the pandemic [2].

The proportion of SARS-CoV-2 infections affecting HCWs reported from a single centre has ranged from 0 to 29% [3–5]. In China approximately 4% of all SARS-CoV-2 infections were in HCWs, with an infection rate three times higher in HCWs compared to the general population [6–8]. Seroprevalence estimates among HCWs have varied between 4% and 19% depending on the jurisdiction [9, 10]. HCWs may be exposed to SARS-CoV-2 in the community, at work from patients as well as fellow HCWs, and may pose a risk to others around them if infected. The World Health Organization (WHO) has identified research priorities related to the burden and risk factors for HCW SARS-CoV-2 infections as well as risk factors for household transmission from HCWs [11]. Our objectives were to describe demographic, exposure, and clinical differences between HCW and non-HCW SARS-CoV-2 infections across all of Ontario, Canada. We further evaluated the risk of death from SARS-CoV-2 between HCWs and non-HCWs, as well as the frequency of HCW household members with SARS-CoV-2 infection.

## Methods

### Design and setting

We conducted a cross-sectional study comparing HCW and non-HCW SARS-CoV-2 infections. Data collection began with the first COVID-19 diagnosed patient in Ontario, Canada on 21 January 2020 until 30 September 2020. We obtained the data from reportable surveillance infectious disease data at Public Health Ontario (PHO). The activities described in this manuscript were conducted in fulfillment of PHO's legislated mandate to provide scientific and technical advice and operational support in an emergency or outbreak situation (*Ontario Agency for Health Protection and Promotion Act, SO 2007, c 10*). Research ethics committee approval was sought from PHO's Ethics Review Board and determined to be not required because the activities described are considered public health practice and not research. Data were fully anonymized and informed consent was not required.

## Data source

We obtained the data from the integrated Public Health Information System (iPHIS), the Toronto Public Health Coronavirus Rapid Entry System (CORES), the Ottawa Public Health COVID-19 Ottawa Database (The COD), Middlesex-London COVID-19 Case and Contact Management tool (CCMtool), and the Public Health Case and Contact Management Solution (CCM) accessed on 14 October 2020 (but only including cases up to 30 September 2020 to account for a delay in reporting). These databases are web-based information systems for the reporting and surveillance of diseases of public health significance in Ontario. PHO is a government corporation dedicated to protecting and promoting the health of all Ontarians and reducing inequities in health. All confirmed SARS-CoV-2 infections are entered by local public health units.

## Definitions and variables

A HCW was defined as an individual who self-reported to have an occupation involving caring for patients including (but not limited to) doctors, nurses, dentists, dental hygienists, midwives, other medical technicians, personal support workers, respiratory therapists, and first responders. The database has additional fields for doctor and nurse, however all other HCWs were recorded as "other" with the opportunity for free text. Data quality was evaluated and HCW cases were excluded if they were <15 or >80 years of age (n = 27). All other individuals with diagnosed SARS-CoV-2 infections were classified as non-HCWs. Demographic information available included gender, age, and comorbidities (anemia, asthma, cancer, cardiovascular condition, chronic liver disease, chronic obstructive pulmonary disease (COPD), diabetes, immunocompromised, neurologic disorder, obesity, pregnancy or 6 weeks post-partum, renal condition, tuberculosis, and other chronic medical condition). Exposures were classified by the local public health unit contact investigation as outbreak associated or close contact to a confirmed or probable SARS-CoV-2 infection, community transmission with no epidemiological link, travel to an endemic area for SARS-CoV-2 outside of Ontario within the incubation period of 14 days, or missing exposure information. A separate variable to identify nosocomial cases was added 3 April 2020. Analysis with the nosocomial variable was limited to this time frame. Clinical symptoms were classified as asymptomatic, presymptomatic (defined as having a testing date prior to symptoms onset date), typical with fever and/or cough, other symptoms, or missing. Clinical outcomes were classified in descending order as died, requiring a ventilator in the intensive care unit, intensive care unit without a ventilator, hospitalized, or not hospitalized.

Onset of illness was defined as symptom onset date, which was available for 69.2% of the cohort. For those missing symptom onset date we calculated the weekly median number of days from test date to symptom onset date where the data was available and performed a deterministic imputation base on the week of testing. If test date was not available we used the weekly median time from symptoms onset to date reported to the local public health unit. Onset date in asymptomatic individuals was the testing date or reporting date if not available.

We defined household spread using a natural language processing algorithm to link confirmed HCWs with SARS-CoV-2 infections to other confirmed household contacts with SARS-CoV-2 infection by residential address. The algorithm used Python's sklearn library. Address text was broken down into short segments (N-grams) with a term-frequency inverse document frequency matrix. The closest proximity match within the matrix was returned and validated using checks for numerical portions of the address field, including suite number if available. The algorithm matched those within the same household or apartment unit, and not to neighbours. If symptom onset dates in the non-HCWs were two or more days earlier than

the HCWs then these were defined as a probable HCW *acquisition*. If household cases symptom onset dates were two or more days following a HCWs' then this was defined as a probable *transmission*. Infections that were -1, 0, or +1 days apart were classified as unknown direction of transmission. As a sensitivity analysis we used ±4 days for greater confidence in the direction of transmission.

## Statistical analysis

Variables were compared between HCWs and non-HCWs by chi-squared tests, t-tests, or non-parametric tests as appropriate in bivariable analyses with two-side p-values <0.05 as statistically significant. Rates of infection in non-HCWs were determined using population denominators from Statistics Canada. HCW denominators were calculated from publicly available sources at the Canadian Institute for Health Information and Statistics Canada [12, 13]. A multivariable logistic regression model was built to evaluate the association between status of HCW (independent variable) and death (dependent variable). Covariates in the model were selected a priori based on their clinical relevance. We included sex (male versus female), age (<30 years, 30–44 years, 60–74 years, or ≥75 years, compared to 45–59 years), comorbidities (asthma, COPD, renal disease, cardiac disease, diabetes, immune compromised or cancer, obesity, or other comorbidities, compared to no comorbidities), working or residing in a long-term care home (yes versus no), and symptoms (fever and/or cough, other symptoms, or missing symptoms compared to asymptomatic). Where necessary missing data was included as its own covariate.

## Results

There are an estimated 552,560 HCWs and 14,311,868 non-HCWs in Ontario. As of 30 September 2020 there were 56,606 confirmed SARS-CoV-2 infections in Ontario, including 7,050 (12.5%) HCWs. There was geographical variability in the proportion of HCW SARS-CoV-2 infections ranging from 2–26% across Ontario's 34 public health units. In general, the regions with the largest numbers of cases did not overlap with those with the highest proportions (Fig 1). For instance, Toronto and Peel regions had the highest numbers of cases (2,211 and 1,148), but were in the lowest quartile of regions by the proportion of cases that were HCWs (11.1% and 11.5%).

HCWs with SARS-CoV-2 infection were more likely to be female and were more commonly between the ages of 30–60 years compared to non-HCWs. There were 11 (0.2%) HCWs ≥75 years of age compared to 8,079 (16.3%) non-HCWs. Approximately 30% of both HCW and non-HCWs had one or more comorbidities, most commonly cardiovascular conditions, diabetes, and asthma (Table 1).

HCWs were identified as cases at a rate 3.7 times higher than non-HCWs with rates of 1,276 per 100,000 compared to 346 per 100,000. There were 161 physicians infected, comprising 2.3% of HCW infections, with an infection rate 1.4 times the general population. There were 1,756 nurses (24.9% of HCWs) who comprised 3.1% of all Ontario infections with an infection rate 3.3 times higher than the general population. There were 5,133 (72.8% of HCWs) other HCWs, including 2,154 personal support workers (PSWs). These HCWs had an infection rate 4.1 times that of the general population (Table 2).

The difference in daily new infection rates varied between 1–7 fold throughout the epidemic. Capacity to test non-hospitalized, non-HCWs, was extremely limited until June 2020 [14]. As testing capacity improved in Ontario the difference in detection rates equilibrated and remained similar through the rise in cases in September 2020 (Fig 2).

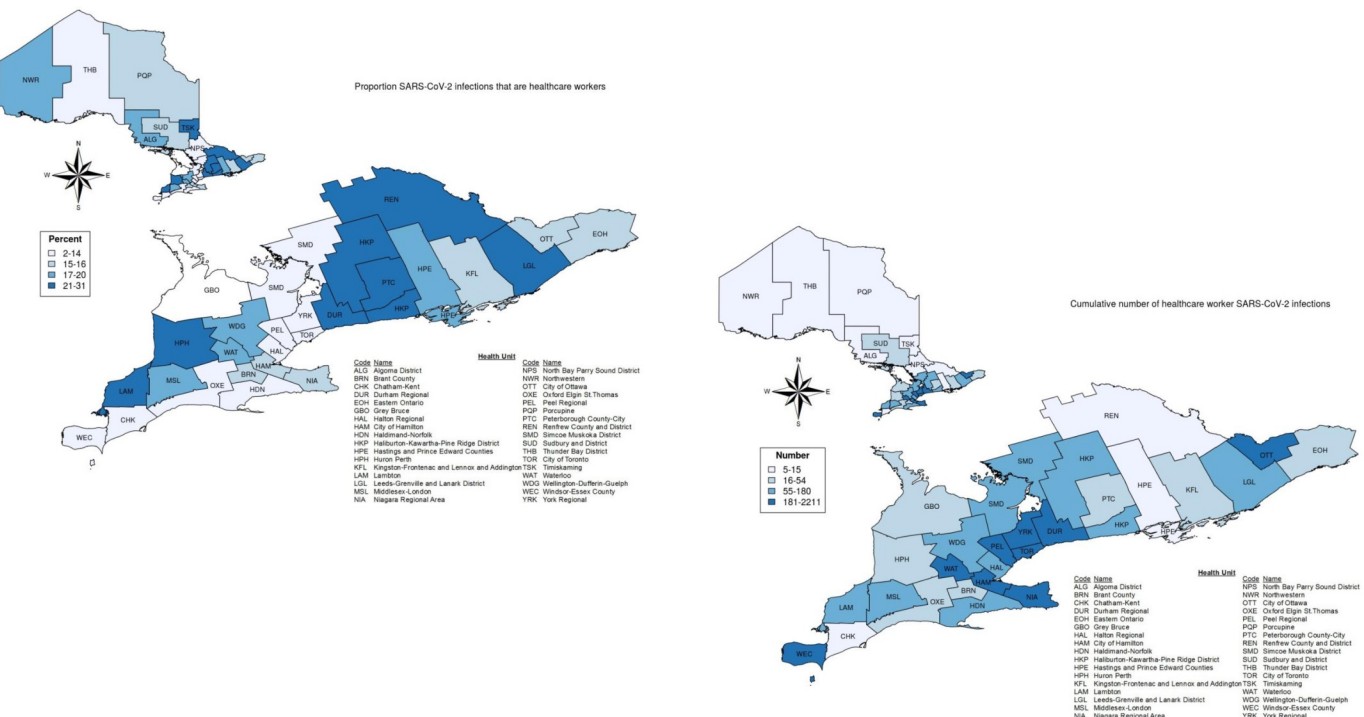

**Fig 1. Geographical variability by Ontario public health unit until 30 September 2020.** Left map shows the percent of total SARS-CoV-2 infections in each region that are healthcare workers. Right map shows the cumulative number of healthcare worker SARS-CoV-2 infections.

There were 1,746 (26.0%) HCW household SARS-CoV-2 infections; of these the median number of cases was 1 with a range of 1–10. We observed that 829 (12.3%) of HCWs probably *transmitted* SARS-CoV-2 to a household member; of these, 20.1% were to children <19 years, 63.7% to those 19–59 years and 16.2% to those ≥60 years. We observed 454 (6.8%) instances where the HCW probably *acquired* the infection from a household contact; of these, 7.5% were from children <19 years, 72.9% from adults 19–59 years and 19.6% from adults ≥60 years. In 463 (6.9%) contacts the direction of transmission could not be determined (Table 3). In the sensitivity analysis using ±4 days, instead of ±2 days, there were 541 (8.1%) *transmissions* and 281 (4.2%) *acquisitions*.

We observed 12 (0.2%) deaths in HCWs; 0 physicians, 1 nurse, 6 PSWs, and 5 unknown, compared to 3,004 (6.1%) in non-HCWs (p<0.001). In the multivariable model HCW status remained strongly associated with a lower risk of death after adjusting for multiple possible confounders including age, sex, comorbidities, long-term care home exposure, and symptoms (adjusted odds ratio (aOR) 0.09; 95%CI 0.05–0.17, p<0.001). Age was strongly associated with death. Comorbidities associated with an increased risk of death were obesity, renal conditions, COPD, immunocompromised state or cancer, and diabetes (p<0.05). Asthma and cardiovascular conditions were not significantly associated with death (Table 4).

## Interpretation

HCWs have comprised 12.5% of the 56,606 confirmed SARS-CoV-2 infections in Ontario, Canada as of 30 September, 2020. The rate of new infections per day varied between one and seven times the general population over time and by type of HCW with physicians being lower risk than nurses, who were lower risk than other specialties combined, which includes PSWs.

**Table 1.  Clinical, demographic and exposure comparison between healthcare workers and non-healthcare workers with SARS-CoV-2 infection (n = 56,606).**

| Variable | Healthcare workers, n (%) | | | | Non-healthcare workers, n (%) | p-value[a] |
|---|---|---|---|---|---|---|
| | Doctor | Nurse | Other[b] | Total | | |
| **Total (% of all SARS-CoV-2 infections)** | **161 (0.3)** | **1756 (3.1)** | **5133 (9.1)** | **7050 (12.5)** | **49556 (87.5)** | |
| **Female gender** | 65 (40.4) | 1516 (86.3) | 4142 (80.7) | 5723 (81.2) | 23479 (47.4) | <0.001 |
| **Age** | | | | | | <0.001 |
| <30 | 25 (15.5) | 443 (25.2) | 890 (17.3) | 1358 (19.3) | 15107 (30.5) | |
| 30–44 | 68 (42.2) | 630 (35.9) | 1603 (31.2) | 2301(32.6) | 10153(20.5) | |
| 45–59 | 36 (22.4) | 557 (31.7) | 2057 (40.1) | 2650 (37.6) | 9581 (19.3) | |
| 60–74 | 29 (18.0) | 125 (7.1) | 576 11.2) | 730 (10.4) | 6630 (13.4) | |
| 75+ | 3 (1.9) | 1 (0.1) | 7 (0.1) | 11 (0.2) | 8079 (16.3) | |
| Unknown | 0 | 0 | 0 | 0 | 6 (<0.1) | |
| **One or more comorbidities[c]** | 36 (22.4) | 537 (30.6) | 1482 (28.9) | 2055 (29.1) | 14073 (28.4) | 0.19 |
| Asthma | 8 (5.0) | 123 (7.0) | 327 (6.4) | 458 (6.5) | 2197 (4.4) | |
| COPD | 0 | 7 (0.4) | 8 (0.2) | 15 (0.2) | 669 (1.3) | |
| Renal conditions | 1 (0.6) | 9 (0.5) | 39 (0.8) | 49 (0.7) | 1020 (2.1) | |
| Cardiovascular conditions | 13 (8.1) | 163 (9.3) | 457 (8.9) | 633 (9.0) | 6012 (12.1) | |
| Diabetes | 5 (3.1) | 108 (6.2) | 364 (7.1) | 477 (6.8) | 3988 (8.0) | |
| Immune compromised or cancer | 3 (1.9) | 45 (2.6) | 132 (2.6) | 180 (2.6) | 1693 (3.4) | |
| Obesity | 1 (0.6) | 43 (2.4) | 89 (1.7) | 133 (1.9) | 589 (1.2) | |
| Other medical conditions | 13 (8.1) | 249 (14.2) | 578 (11.3) | 840 (11.9) | 6274 (12.7) | |
| **Clinical presentation** | | | | | | |
| Presymptomatic | 1 (0.6) | 30 (1.7) | 77 (1.5) | 108 (1.5) | 601 (1.2) | 0.02 |
| **Presenting symptoms** | | | | | | <0.001 |
| Asymptomatic | 19 (11.8) | 177 (10.1) | 951 (18.5) | 1147 (16.3) | 8680 (17.5) | |
| Fever and/or cough | 90 (55.9) | 981 (55.9) | 2578 (50.2) | 3802 (53.9) | 23165 (46.7) | |
| Other symptoms | 32 (19.9) | 427 (24.3) | 1359 (26.5) | 1906 (27.0) | 12918 (26.1) | |
| Missing symptom data | 4 (2.5) | 40 (2.3) | 220 (4.3) | 195 (2.8) | 4793 (9.7) | |
| **Outcomes** | | | | | | <0.001 |
| Not hospitalized | 150(93.2) | 1687 (96.1) | 4966 (96.7) | 6803 (96.5) | 42713 (86.2) | |
| Hospitalized | 7 (4.3) | 51 (2.9) | 121 (2.4) | 179 (2.5) | 3128 (6.3) | |
| Intensive care unit—not on ventilator | 1 (0.6) | 10 (0.6) | 30 (0.6) | 41 (0.6) | 500 (1.0) | |
| Intensive care unit–on a ventilator | 3 (1.9) | 7 (0.4) | 5 (0.1) | 15 (0.2) | 211 (0.4) | |
| Died | 0 | 1 (0.1) | 11 (0.2) | 12 (0.2) | 3004 (6.1) | |
| **Resolved** | 161 (100.0) | 1752 (99.8) | 5109 (99.5) | 7022 (99.6) | 45737 (92.3) | <0.001 |
| **Exposure history** | | | | | | <0.001 |
| Outbreak-associated or close contact of a confirmed case | 67 (41.6) | 1347 (76.7) | 3956 (77.1) | 5370 (76.2) | 31268 (63.1) | |
| No known epidemiological link | 57 (35.4) | 291 (16.6) | 835 (16.3) | 1183 (16.8) | 8281 (16.7) | |
| Travel-Related | 28 (17.4) | 77 (4.4) | 152 (3.0) | 257 (3.6) | 1997 (4.0) | |
| Information missing | 9 (5.6) | 41 (2.3) | 190 (3.7) | 240 (3.4) | 8010 (16.2) | |
| **Exposed to long-term care home** | 6 (3.7) | 586 (33.4) | 2334 (45.5) | 2926 (41.5) | 6926 (14.0) | |
| **Nosocomial transmission[d]** | | | | | | <0.001 |
| Yes | 1 (0.9) | 80 (5.1) | 106 (2.3) | 187 (3.0) | 410 (0.9) | |
| No | 33 (30.3) | 301 (19.0) | 820 (18.0) | 1154 (18.4) | 6523 (14.1) | |

(*Continued*)

**Table 1.** (Continued)

| Variable | Healthcare workers, n (%) | | | | Non-healthcare workers, n (%) | p-value[a] |
|---|---|---|---|---|---|---|
| | Doctor | Nurse | Other[b] | Total | | |
| Unknown or missing | 75 (68.8) | 1201 (75.9) | 3642 (79.7) | 4918 (78.6) | 39490 (85.1) | |

[a] Statistical comparison between total healthcare workers and non-healthcare workers

[b] 2154 (30.6%) were personal support workers identified through free text searching.

[c] Comorbidities include; anemia or hemoglobinopathy, asthma, cancer, cardiovascular disease, chronic liver disease, chronic obstructive pulmonary disease (COPD), diabetes, immunocompromised, neurological disorder, obesity, pregnant or post-partum, renal conditions, tuberculosis, other chronic medical condition.

[d] Nosocomial variable added on April 3, 2020, therefore the denominators used were 6259 healthcare workers (109 doctors, 1582 nurses, and 4568 other) and 46423 non-healthcare workers.

Interestingly, during the period of time without restrictions in access to testing (June to September 2020) for non-HCWs there were small to no differences between the rate of HCW and non-HCW SARS-CoV-2 infections.

Data from China suggest that approximately 4% of all SARS-CoV-2 infections were in HCWs [7, 8]. In the United States 6% of hospitalized SARS-CoV-2 infections were HCWs [15]. There has been wide variation in single centre reports of HCW COVID-19 infections. Early in the pandemic one report from Wuhan observed that 29% of 138 hospitalized SARS-CoV-2 infections were in HCWs, with at least 10 of those related to a possible super-spreader event [5]. However, other centres have reported no SARS-CoV-2 infections despite significant exposures [3, 16]. In a long term care facility outbreak in Washington state, HCWs comprised 26% of infections, however the clinical course was substantially less severe in HCWs compared

**Table 2.** Rate of new SARS-CoV-2 infections in healthcare workers and non-healthcare workers.

| | Healthcare workers | | Non-healthcare workers | |
|---|---|---|---|---|
| | Number (%) | Rate per 100,000 | Number (%) | Rate per 100,000 |
| **Total** | 7050 | 1275.9 | 49556 | 346.3 |
| Physicians | 161 (2.3) | 475.3 | NA | NA |
| Nurses | 1756 (24.9) | 1128.8 | NA | NA |
| Other | 5133 (72.8) | 1413.6 | NA | NA |
| **Average daily new infections** | | | | |
| February 15–29 | 3 (<0.1) | 0.5 | 28 (0.1) | 0.2 |
| March 1–14 | 82 (1.2) | 14.8 | 645 (1.3) | 4.5 |
| March 15–31 | 1001 (14.2) | 181.2 | 3840 (7.8) | 26.8 |
| April 1–14 | 1456 (20.7) | 263.5 | 5338 (10.8) | 37.3 |
| April 15–30 | 1720 (24.4) | 311.3 | 6070 (12.3) | 42.4 |
| May 1–14 | 783 (11.1) | 141.7 | 3827 (7.7) | 26.7 |
| May 15–31 | 593 (8.4) | 107.3 | 5178 (10.5) | 36.2 |
| June 1–14 | 260 (3.7) | 47.1 | 2497 (5.0) | 17.4 |
| June 15–30 | 228 (3.2) | 41.3 | 2376 (4.8) | 16.6 |
| July 1–14 | 118 (1.7) | 21.4 | 1694 (3.4) | 11.8 |
| July 15–31 | 111 (1.6) | 20.1 | 1923 (3.9) | 13.4 |
| August 1–14 | 68 (1.0) | 12.3 | 1208 (2.4) | 8.4 |
| August 15–31 | 106 (1.5) | 19.2 | 1936 (3.9) | 13.5 |
| September 1–14 | 149 (2.1) | 27.0 | 3602 (7.3) | 25.2 |
| September 15–30 | 372 (5.3) | 67.3 | 9381 (18.9) | 65.5 |

NA = not applicable.

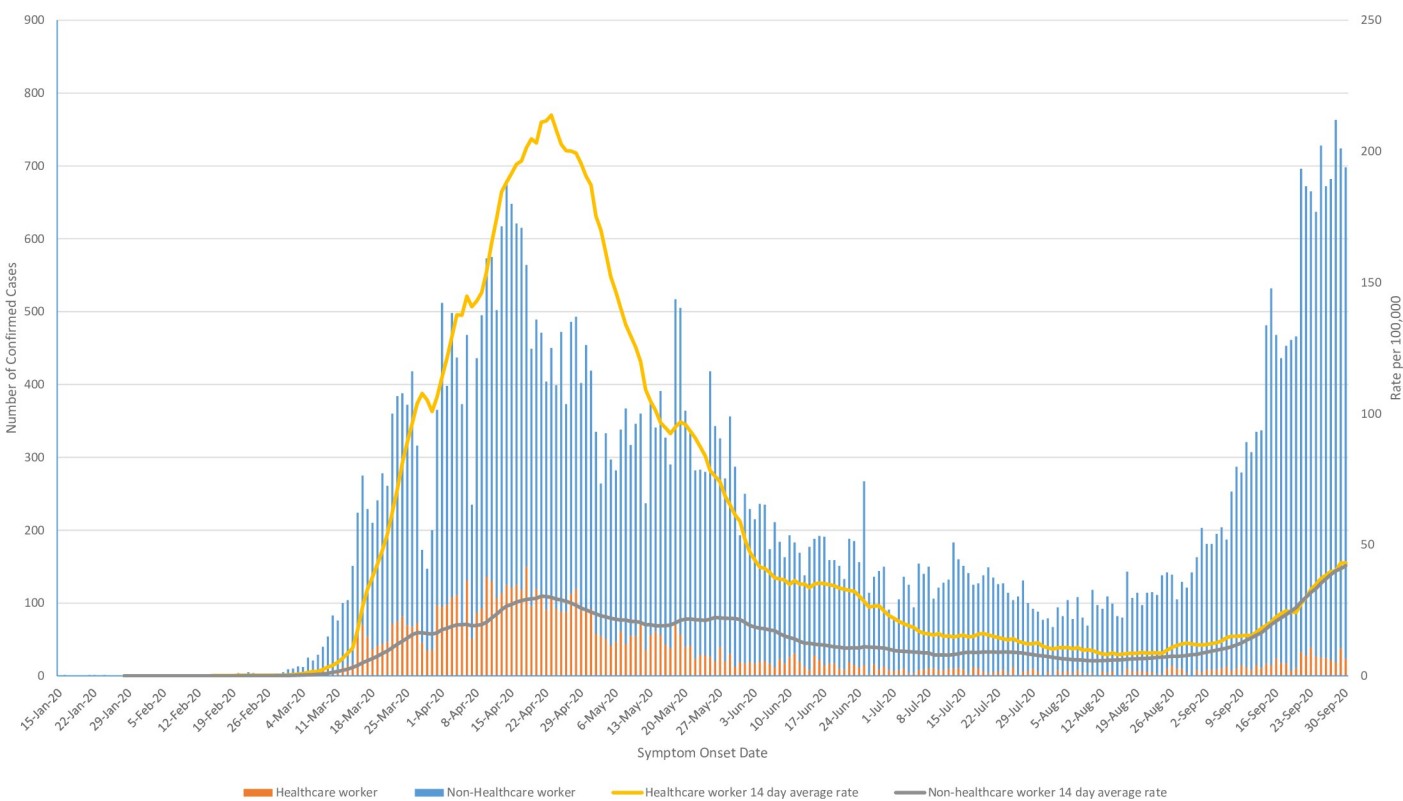

**Fig 2.** Epidemic curve by symptom onset date showing daily new case numbers until 30 September 2020 (left axis) and 14-day moving average of daily rate of new SARS-CoV-2 infections by symptom onset date (right axis) for healthcare workers and non-healthcare workers.

to residents and visitors [4]. Seropositivity studies have demonstrated wide variability from 4% to 19% of HCWs have detectable SARS-CoV-2 IgG antibodies [9, 10, 17]. This variability is likely related to differences in community transmission rates and personal protective equipment (PPE) availability. Some studies have demonstrated similar infection prevalence among clinical and non-clinical facing staff suggesting that a substantial proportion of HCW infections were likely acquired from the community or co-workers [18–20].

The Center for Disease Control and Prevention in the United States reported on over 9,000 infected HCWs. Compared to our study they observed higher percent of hospitalizations (10% versus 2.5% in our study) but similar rate of death (0.3% versus 0.2%) [21]. In a study from Mexico, 13.1% of all SARS-CoV-2 infections were HCWs, and the risk of death was lower (9.9% in non-HCWs compared to 1.9% in HCWs) which persisted after multivariable adjustment for age, sex, and comorbidities (OR 0.53; 95%CI 0.46–0.61) [22]. In Mexico, there were

**Table 3. Healthcare worker within household SARS-CoV-2 acquisitions and transmissions (n = 6,716 healthcare workers).**

| | Number (%) | Age group of household contact, n (row %) | | |
|---|---|---|---|---|
| | | **<19 years** | **19–59 years** | **≥60 years** |
| **HCW acquired from household contact** | 454 (6.8) | 34 (7.5) | 331 (72.9) | 89 (19.6) |
| **HCW transmitted to household contact** | 829 (12.3) | 167 (20.1) | 528 (63.7) | 134 (16.2) |
| **Household infection with unknown direction of transmission** | 463 (6.9) | 74 (16.0) | 315 (68.0) | 74 (24.9) |
| **Total household transmissions involving HCWs** | 1746 (26.0) | 275 (15.8) | 1174 (67.2) | 297 (17.0) |

HCW = healthcare worker; 334 HCWs were missing a home address or were linked to a congregate setting and excluded from this analysis.

**Table 4. Multivariable logistic regression model evaluating risk of death subsequent to SARS-CoV-2 infection.**

| Variable | Adjusted Odds Ratio | 95% Confidence Intervals | p-value |
|---|---|---|---|
| Healthcare worker | | | |
| Yes | 0.09 | 0.05–0.165 | <0.001 |
| No | Reference | - | - |
| Age | | | |
| <30 years | 0.03 | 0.01–0.08 | <0.001 |
| 30–44 years | 0.132 | 0.08–0.23 | <0.001 |
| 45–59 years | Reference | - | - |
| 60–75 years | 5.47 | 4.44–6.72 | <0.001 |
| ≥75 years | 20.41 | 16.68–24.97 | <0.001 |
| Comorbidities | | | |
| Asthma | 0.85 | 0.66–1.09 | 0.198 |
| COPD | 1.39 | 1.14–1.70 | 0.001 |
| Renal conditions | 1.61 | 1.35–1.92 | <0.001 |
| Cardiovascular conditions | 1.10 | 0.99–1.22 | 0.078 |
| Diabetes | 1.13 | 1.01–1.28 | 0.042 |
| Immune compromise or cancer | 1.48 | 1.27–1.74 | <0.001 |
| Obesity | 1.66 | 1.19–2.30 | 0.003 |
| Other medical conditions | 1.18 | 1.06–1.31 | 0.003 |
| None | Reference | - | - |
| Exposed to long-term care home | | | |
| Yes | 3.04 | 2.75–3.37 | <0.001 |
| No | Reference | - | - |
| Symptoms | | | |
| Fever and/or cough | 3.95 | 3.42–4.57 | <0.001 |
| Other symptoms | 3.60 | 3.06–4.23 | <0.001 |
| Missing | 3.53 | 2.98–4.18 | <0.001 |
| Asymptomatic | Reference | - | - |

COPD = chronic obstructive pulmonary disease.

116 physician deaths among the 4,609 infections (2.5%), which was significantly higher compared to the 33 deaths among 6,240 nurses (0.5%, p<0.001). In our study we observed no physician deaths and 1 nurse death among 161 (0%) and 1,752 (0.05%) cases, respectively.

Studies from China and the Netherlands reported that approximately 1% of HCWs within hospitals were infected with SARS-CoV-2 earlier in the pandemic [18, 23]. In our study, 1.3% of all HCWs and 0.3% of non-HCWs in Ontario have been diagnosed with SARS-CoV-2 infection, including 0.5% of doctors and 1.1% of nurses. A report from Alberta, Canada identified that 0.1% of all HCWs had SARS-CoV-2 infections (including 0.3% of doctors) compared to 0.1% of the general population [24]. Alberta has similar IPAC guidance on PPE to Ontario which includes surgical masks, eye protection, gloves, and gowns for suspected or confirmed COVID-19 patients and N95 respirators for aerosol generating medical procedures. Our study was not able to evaluate the adequacy of PPE used by HCWs, however it is unlikely to explain the differences between Ontario and Alberta. Ontario likely has substantially more undiagnosed cases in the general population as well as more long-term care home outbreaks. The most important risk factors previously reported for HCW SARS-CoV-2 infections are community (i.e. restaurants, bars, and public transportation) and household exposures [25]. Further study is needed to better understand the nosocomial risk for SARS-CoV-2 infection among HCWs.

We observed a 10-fold lower odds of death among HCWs compared to non-HCWs even after adjustment for important confounders of age, sex, symptoms, and comorbidities. Testing bias, particularly earlier in the pandemic, likely contributed to the discrepancy in HCW and non-HCW rates which is supported by our epidemic curve in Fig 2. Improved access to PPE and adherence to infection prevention and control guidance is also possible. HCWs have been a priority population for testing throughout the pandemic, even during times of limited capacity. Between June and September 2020 mobility restrictions were gradually lifted in the province and there was substantially improved access to testing [14]. This finding suggests there were substantial underestimates of true population disease burden, which may actually be closer to the rate we observed in HCWs. In a report from Lombardy, Italy that performed serological studies on HCWs and non-HCWs, they identified that 23% of HCWs sampled were positive for COVID-19 antibodies compared to 62% of the general population [26].

HCWs may be a potential source of SARS-COV-2 transmission to both patients and household contacts. In particular since presymptomatic cases comprise a substantial portion of transmissions [27]. Transmitting to household members has been a source of stress for HCWs [28] and has been identified as a knowledge gap by the WHO [11]. We observed that 12.3% of infected HCWs likely transmitted SARS-CoV-2 to a household contact. Using a stricter definition of 4 days, we observed a transmission rate of 8.1%. Interestingly, children were relatively infrequently the source of infection compared to becoming secondary household cases (7.5% versus 20.1%). This finding may be due to the lower risk of SARS-CoV-2 infections in children earlier in the pandemic or children may be inefficient transmitters of SARS-CoV-2 [29].

This study has some limitations. The data quality is dependent on entry by 34 public health units across Ontario. Data completeness and quality may vary and it is possible some HCWs were misclassified. The variable for nosocomial transmission was added later in the study period and largely incomplete even after restricting to the later time period, therefore limited inference can be drawn. This may be in part due to the challenge of assigning causality of infections without a clear exposure history or multiple potential exposures. The general population, doctor, and nurse denominators used to calculate infection rates are accurate and current, however there is no complete data source for other HCWs in Ontario. We used various sources to arrive at an overall estimated HCW denominator which may impact the ability to compare rates. We lacked granularity in the type of HCW beyond physician or nurse. We searched the free text for PSW, however as a result we likely under captured this group. We quantified household transmissions using a natural language processing algorithm, however we do not have data on the denominator of household contacts to estimate attack rates. Finally, while HCWs were priority groups for testing throughout the pandemic, testing criteria and capacity varied substantially. Case identification was substantially better throughout for HCWs and as a result general population cases are undoubtedly underestimates.

In conclusion, 1.3% of HCWs in Ontario have been diagnosed with SARS-CoV-2 compared to 0.3% of non-HCWs after the first 8 months of the COVID-19 pandemic. Substantial under diagnosis of non-HCWs earlier in the pandemic at least partially explains this discrepancy. We observed only 12 HCW deaths, and HCW lethality remained significantly lower compared to non-HCWs after multivariable adjustment.

## Acknowledgments

We would like to thank James Johnson for developing the natural language processing algorithm linking household cases, as well as Brendan Smith and Christine Warren for their help determining total numbers of healthcare workers in Ontario.

## Author Contributions

**Conceptualization:** Kevin L. Schwartz, Sarah A. Buchan, Kevin A. Brown, Gary Garber.

**Formal analysis:** Camille Achonu.

**Methodology:** Kevin L. Schwartz, Camille Achonu, Sarah A. Buchan, Kevin A. Brown, Brenda Lee, Michael Whelan, Julie HC Wu, Gary Garber.

**Project administration:** Julie HC Wu.

**Supervision:** Gary Garber.

**Writing – original draft:** Kevin L. Schwartz.

**Writing – review & editing:** Camille Achonu, Sarah A. Buchan, Kevin A. Brown, Brenda Lee, Michael Whelan, Julie HC Wu, Gary Garber.

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
