## [Decision Letter · Decision Letter 0]

7 Aug 2020

PONE-D-20-18932

COVID-19 infections among Healthcare Workers and Transmission within Households

PLOS ONE

Dear Dr. Schwartz,

Thank you for submitting your manuscript to PLOS ONE. After careful consideration, we feel that it has merit but does not fully meet PLOS ONE’s publication criteria as it currently stands. Therefore, we invite you to submit a revised version of the manuscript that addresses the points raised during the review process.

In addition to the minor issues raised by the reviewer, there are several major issues that need to be addressed.

1.

Throughout the manuscript, COVID-19 cases (disease) have been confused with SARS-CoV-2 infections. This is clear because also the asymptomatic subjects have been included among the "cases". This is plain wrong, misleading for the reader, and definitely requires to be corrected everywhere.

2.

In the methodology, it is written that all COVID-19 cases (which is referred to SARS-CoV-2 infections) have been captured by Public Health Ontario. Later in the manuscript, it is written repeatedly that many infections have likely been lost. Consistently, the authors should refer to their sample as "all the SARS-CoV-2 infections that have been diagnosed".

3.

No details have been provided on the "natural language processing" algorithm, that has been used to identify the household transmission (an essential part of the study). Was the household referred solely to those residing in the same house, or also to the neighbours? This clearly needs to be clarified.

4.

Mortality has been sometimes confused with lethality. Please correct it everywhere.

5.

Insufficient details have been provided on the definition of healthcare workers. It is only written that they are "caring for patients". Does this definition include also cohabiting caregivers who are not employees of the National healthcare system? If so, how many? Given that only 22.6% of the healthcare workers were physicians or nurses, more information on the remaining 77.4% is definitively needed. In the manuscript, the only detail is given in the footnote b of Table 1, where it is written that "at least 708 (21.6%) were personal support workers identified through free text searching". First, what does "at least mean? Second, what about the remaining 1968? Who are they? This is an essential information that is needed to understand the adopted definition of healthcare worker.

6.

A number of comorbidities have been listed as available, but these data have not been reported and, even more importantly, they have not been used to perform multivariate analyses, which are essential to make any meaningful comparison between the risk of death of healthcare workers and the rest of the population. This point needs to be clarified and data, if really available, provided for both healthcare workers and the rest of the population. Also, if these data are available, a multivariate analysis predicting the risk of death of healthcare workers versus others should be performed and would be the most interesting part of the study (for the limitations that will be mentioned later).

7.

The information on nosocomial transmission is missing for 76.9% of the healthcare workers, and 87.5% of the population. Despite this clear, huge bias, the authors extensively discuss these data, just briefly mentioning the missing data issue. With such a large amount of missing data, any discussion of the result should be avoided. The risk of bias of these data is too large to be meaningful in any way, and they can also be misleading. All analyses on nosocomial infections can be shown but briefly mentioned in the Results as merely indicative. Do not mention them in the Discussion (if not in the limitations), Conclusions or Abstract.

8.

Both the Results and the Discussion need to be improved. In the Results there some confusion, the most important data, those on the comparisons of the infection and mortality rates, have been reported after some trivial data (as the difference in the proportion of asymptomatic cases, a clear example of a statistically significant yet meaningless finding - 8.1% versus 7.0%).

The Discussion is redundant, and does not follow a clear reasoning. There also are some speculations (page 8, lines 177-178 on Ontario), some sentences that are to placed into an Introduction (page 10, lines 212-215) and, most importantly, it is totally unclear how the conclusions of the authors "we feel that data highlight the importance of community and household risk for HCWs, maintaining physicial distancing from colleagues and utilizing addition PPI..."

First, all the conclusions were known before the study. Second, and most importantly, it is not explained why the results of the study lead the authors to "feel" this way. In this study, nosocomial infections (with all the limitations above reported) were as low as 3.6%, and household transmissions were also relatively low (9.8%; lower than many other non-cited studies). It is unclear why these findings should bring us to "emphasize the protection policies of healthcare workers". This definitively needs explanation.

Also, as briefly mentioned before, the discussion on the potential explanations for the observed difference in lethality (not mortality) should be based upon multivariate analyses. Otherwise, it is speculative.

9.

Page 10, lines 217-220: how many children were there to be infected? This point needs clarification.

10.

Limitations need to be expanded, including the absence of a multivariate analysis, the huge missing data (not only for nosocomial infections, but also the symptoms were missing for almost 40% of the population), and the unavoidable underestimation of the SARS-CoV-2 infection in the total population.

11.

The abstract is the weakest part of the manuscript and needs to be extensively revised. First, the comparison is not stated, it is repeatedly stated that "HCWs were more likely...", but it is not clear "than who?".

12.

No definition of acquisition is provided, nosocomial infections need to be cut, mortality is lethality, COVID-19 is SARS-CoV-2 infection, it is not clear whether the 9.8% of probable secondary household transmissions are cases or episodes, low numbers are low percentages, the conclusion is unsubstantiated, and other issues as above.

Minor issues

13.

References 4-6 could be updated.

14.

Page 3, lines 47-48: it is unclear whether this point is related to the aim of the study.

We look forward to receiving your revised manuscript.

Kind regards,

Lamberto Manzoli, M.D., M.P.H.

Academic Editor

PLOS ONE

Journal Requirements:

2. In your ethics statement in the Methods section and in the online submission form, please clarify whether all data were fully anonymized before you accessed them and/or whether the IRB or ethics committee waived the requirement for informed consent.

4. We note that Figure 1 in your submission contain map images which may be copyrighted. All PLOS content is published under the Creative Commons Attribution License (CC BY 4.0), which means that the manuscript, images, and Supporting Information files will be freely available online, and any third party is permitted to access, download, copy, distribute, and use these materials in any way, even commercially, with proper attribution. For these reasons, we cannot publish previously copyrighted maps or satellite images created using proprietary data, such as Google software (Google Maps, Street View, and Earth). For more information, see our copyright guidelines: http://journals.plos.org/plosone/s/licenses-and-copyright.

4.1.    You may seek permission from the original copyright holder of Figure 1 to publish the content specifically under the CC BY 4.0 license.

4.2.    If you are unable to obtain permission from the original copyright holder to publish these figures under the CC BY 4.0 license or if the copyright holder’s requirements are incompatible with the CC BY 4.0 license, please either i) remove the figure or ii) supply a replacement figure that complies with the CC BY 4.0 license. Please check copyright information on all replacement figures and update the figure caption with source information. If applicable, please specify in the figure caption text when a figure is similar but not identical to the original image and is therefore for illustrative purposes only.

Reviewers' comments:

Reviewer's Responses to Questions

**Comments to the Author**

1. Is the manuscript technically sound, and do the data support the conclusions?

Reviewer #1: Yes

2. Has the statistical analysis been performed appropriately and rigorously? 

Reviewer #1: Yes

3. Have the authors made all data underlying the findings in their manuscript fully available?

Reviewer #1: Yes

4. Is the manuscript presented in an intelligible fashion and written in standard English?

Reviewer #1: Yes

5. Review Comments to the Author

Reviewer #1: I have only a few minor suggestions:

p. 2 line 29 Given that the majority of HCW are ‘other’, it would be helpful to include at least the example of the largest group in the abstract. If these are personal support workers, I think this is important to put in the abstract.

p. 3 line 49 suggest revising to read “reported from a single centre has ranged”

p. 4 line 75. Many people will not know what a Crown corporation is. Can you use a more descriptive or generic term?

p. 4 line 81 insert ‘and’ before comorbidities

p. 5 line 100. Is there any way or sense of how multiple occupancy addresses (apartment buildings were included and how they were handled in the data analysis?

p. 7 and other pages. Can you clarify if you are defining nosocomial and acquired in a hospital or acquired in the place the HCW works/from a patient. As most infections are in personal support workers, I found this confusing and the discuss about this on p.8 line 178 to the end of the paragraph took me several readings to understand and I am still not sure I am following correctly. Is all of this discussion about the fact that it is hard to be sure how HCWs were infected, or is it more complex?

6. PLOS authors have the option to publish the peer review history of their article (what does this mean?). If published, this will include your full peer review and any attached files.

Reviewer #1: **Yes: **Annette M. Totten

---

## [Author Response · Author response to Decision Letter 0]

26 Nov 2020

1.

Throughout the manuscript, COVID-19 cases (disease) have been confused with SARS-CoV-2 infections. This is clear because also the asymptomatic subjects have been included among the "cases". This is plain wrong, misleading for the reader, and definitely requires to be corrected everywhere.

Response: This has been corrected throughout.

2.

In the methodology, it is written that all COVID-19 cases (which is referred to SARS-CoV-2 infections) have been captured by Public Health Ontario. Later in the manuscript, it is written repeatedly that many infections have likely been lost. Consistently, the authors should refer to their sample as "all the SARS-CoV-2 infections that have been diagnosed".

Response: This has been corrected in the methods and throughout.

3.

No details have been provided on the "natural language processing" algorithm, that has been used to identify the household transmission (an essential part of the study). Was the household referred solely to those residing in the same house, or also to the neighbours? This clearly needs to be clarified.

Response: Further details of the NLP algorithm added (page 6-7 line 127-131): “The algorithm used Python’s sklearn library. Address text was broken down into short segments (N-grams) with a term-frequency inverse document frequency matrix. The closest proximity match within the matrix was returned and validated using checks for numerical portions of the address field, including suite number if available. The algorithm matched those within the same household or apartment unit, and not to neighbours.”

4.

Mortality has been sometimes confused with lethality. Please correct it everywhere.

Response: The term “mortality” has been changed to “death” or “lethality” throughout.

5.

Insufficient details have been provided on the definition of healthcare workers. It is only written that they are "caring for patients". Does this definition include also cohabiting caregivers who are not employees of the National healthcare system? If so, how many? Given that only 22.6% of the healthcare workers were physicians or nurses, more information on the remaining 77.4% is definitively needed. In the manuscript, the only detail is given in the footnote b of Table 1, where it is written that "at least 708 (21.6%) were personal support workers identified through free text searching". First, what does "at least mean? Second, what about the remaining 1968? Who are they? This is an essential information that is needed to understand the adopted definition of healthcare worker.

Response: Additional details on the definition of HCW has been added (Page 5 line 100-104). “A HCW was defined as an individual who self-reported to have an occupation involving caring for patients including (but not limited to) doctors, nurses, dentists, dental hygienists, midwives, other medical technicians, personal support workers, respiratory therapists, and first responders. The database has fields for doctor and nurse, however all other HCWs were recorded as “other” with the opportunity for free text.”

6.

A number of comorbidities have been listed as available, but these data have not been reported and, even more importantly, they have not been used to perform multivariate analyses, which are essential to make any meaningful comparison between the risk of death of healthcare workers and the rest of the population. This point needs to be clarified and data, if really available, provided for both healthcare workers and the rest of the population. Also, if these data are available, a multivariate analysis predicting the risk of death of healthcare workers versus others should be performed and would be the most interesting part of the study (for the limitations that will be mentioned later).

Response: A multivariable model evaluating the risk of death between HCWs and non-HCWs has now been added as suggested (Table 4). In addition, the comorbidity data has been expanded and utilized for this analysis as suggested.

7.

The information on nosocomial transmission is missing for 76.9% of the healthcare workers, and 87.5% of the population. Despite this clear, huge bias, the authors extensively discuss these data, just briefly mentioning the missing data issue. With such a large amount of missing data, any discussion of the result should be avoided. The risk of bias of these data is too large to be meaningful in any way, and they can also be misleading. All analyses on nosocomial infections can be shown but briefly mentioned in the Results as merely indicative. Do not mention them in the Discussion (if not in the limitations), Conclusions or Abstract.

Response: Reference to the nosocomial data in discussion, conclusion, and abstract has been removed (aside from the noted limitation on line 300).

8.

Both the Results and the Discussion need to be improved. In the Results there some confusion, the most important data, those on the comparisons of the infection and mortality rates, have been reported after some trivial data (as the difference in the proportion of asymptomatic cases, a clear example of a statistically significant yet meaningless finding - 8.1% versus 7.0%).

The Discussion is redundant, and does not follow a clear reasoning. There also are some speculations (page 8, lines 177-178 on Ontario), some sentences that are to placed into an Introduction (page 10, lines 212-215) and, most importantly, it is totally unclear how the conclusions of the authors "we feel that data highlight the importance of community and household risk for HCWs, maintaining physicial distancing from colleagues and utilizing addition PPI..."

First, all the conclusions were known before the study. Second, and most importantly, it is not explained why the results of the study lead the authors to "feel" this way. In this study, nosocomial infections (with all the limitations above reported) were as low as 3.6%, and household transmissions were also relatively low (9.8%; lower than many other non-cited studies). It is unclear why these findings should bring us to "emphasize the protection policies of healthcare workers". This definitively needs explanation.

Also, as briefly mentioned before, the discussion on the potential explanations for the observed difference in lethality (not mortality) should be based upon multivariate analyses. Otherwise, it is speculative.

Response: The results and discussion has been updated with the new results and substantially re-written with sections removed and/or revised as suggested. We have added the multivariable model with death as the outcome (table 4) and added discussion on this. We hope you will find the discussion more organized and supported by the data presented.

9.

Page 10, lines 217-220: how many children were there to be infected? This point needs clarification.

Response: Denominator for household contacts are not known. As stated in the methods this data is only positive cases with SARS-CoV-2 infection. This has been added to the limitations (line 308)

10.

Limitations need to be expanded, including the absence of a multivariate analysis, the huge missing data (not only for nosocomial infections, but also the symptoms were missing for almost 40% of the population), and the unavoidable underestimation of the SARS-CoV-2 infection in the total population.

Response: A multivariable model was added and reference to nosocomial data removed due to the large amount of missing data. With updating the results missing symptom data has been greatly reduced. The missing data has been retained in the multivariable model as its own covariate. Further discussion throughout the manuscript has been added related to the underestimation of general population cases.

11.

The abstract is the weakest part of the manuscript and needs to be extensively revised. First, the comparison is not stated, it is repeatedly stated that "HCWs were more likely...", but it is not clear "than who?".

Response: The abstract has been re-written and updated with the new results.

12.

No definition of acquisition is provided, nosocomial infections need to be cut, mortality is lethality, COVID-19 is SARS-CoV-2 infection, it is not clear whether the 9.8% of probable secondary household transmissions are cases or episodes, low numbers are low percentages, the conclusion is unsubstantiated, and other issues as above.

 Response: Acquisition definition is provided in the methods lines 131-135 (household case with symptom onset data 2+ days prior to the HCW). As a sensitivity analysis we used +/-4 days. Reference to nosocomial variable has been removed from the results text and discussion (aside from the limitations). Mortality has been changed to death or lethality throughout as suggested. COVID-19 has been changed to SARS-CoV-2 infection as suggested throughout. The household transmission data represents new infections (see methods lines 131-135). The results have been substantially revised to not discuss less important findings. The discussion and conclusion (including in the abstract) have been substantially revised and re-written based on the feedback provided and updated results. 

Minor issues

13.

References 4-6 could be updated.

Response: Newer references have been added in addition to these from earlier in the pandemic.

14.

Page 3, lines 47-48: it is unclear whether this point is related to the aim of the study.

Response: This line has been removed.

We look forward to receiving your revised manuscript.

Kind regards,

Lamberto Manzoli, M.D., M.P.H.

Academic Editor

PLOS ONE

Journal Requirements:

2. In your ethics statement in the Methods section and in the online submission form, please clarify whether all data were fully anonymized before you accessed them and/or whether the IRB or ethics committee waived the requirement for informed consent.

 Response: This statement was added (Page 4 line 70).

Response: Public Health Ontario (PHO) cannot disclose the underlying data. Doing so would compromise individual privacy contrary to PHO’s ethical and legal obligations. Restricted access to the data may be available under conditions prescribed by the Ontario Personal Health Information Protection Act, 2004, the Ontario Freedom of Information and Protection of Privacy Act, the Tri-Council Policy Statement: Ethical Conduct for Research Involving Humans (TCPS 2 (2018)), and PHO privacy and ethics policies. Data are available for researchers who meet PHO’s criteria for access to confidential data. Information about PHO’s data access request process is available on-line at https://www.publichealthontario.ca/en/data-and-analysis/using-data/data-requests. 

4. We note that Figure 1 in your submission contain map images which may be copyrighted. All PLOS content is published under the Creative Commons Attribution License (CC BY 4.0), which means that the manuscript, images, and Supporting Information files will be freely available online, and any third party is permitted to access, download, copy, distribute, and use these materials in any way, even commercially, with proper attribution. For these reasons, we cannot publish previously copyrighted maps or satellite images created using proprietary data, such as Google software (Google Maps, Street View, and Earth). For more information, see our copyright guidelines: http://journals.plos.org/plosone/s/licenses-and-copyright.

4.1. You may seek permission from the original copyright holder of Figure 1 to publish the content specifically under the CC BY 4.0 license.

4.2. If you are unable to obtain permission from the original copyright holder to publish these figures under the CC BY 4.0 license or if the copyright holder’s requirements are incompatible with the CC BY 4.0 license, please either i) remove the figure or ii) supply a replacement figure that complies with the CC BY 4.0 license. Please check copyright information on all replacement figures and update the figure caption with source information. If applicable, please specify in the figure caption text when a figure is similar but not identical to the original image and is therefore for illustrative purposes only.

Response: The figures are not copyrighted and were created by the authors.

Reviewers' comments:

Reviewer's Responses to Questions

Comments to the Author

1. Is the manuscript technically sound, and do the data support the conclusions?

Reviewer #1: Yes

2. Has the statistical analysis been performed appropriately and rigorously? 

Reviewer #1: Yes

3. Have the authors made all data underlying the findings in their manuscript fully available?

Reviewer #1: Yes

4. Is the manuscript presented in an intelligible fashion and written in standard English?

Reviewer #1: Yes

5. Review Comments to the Author

Reviewer #1: I have only a few minor suggestions:

p. 2 line 29 Given that the majority of HCW are ‘other’, it would be helpful to include at least the example of the largest group in the abstract. If these are personal support workers, I think this is important to put in the abstract.

Response: We have further explained the definition and data available to describe the type of HCW (page 5 line 100-104): “A HCW was defined as an individual who self-reported to have an occupation involving caring for patients including (but not limited to) doctor, nurse, dentist, dental hygienist, midwife, other medical technicians, personal support worker, respiratory therapist, and first responder. The database has fields for doctor and nurse, however all other HCWs were recorded as “other” with the opportunity for free text.” From the data we cannot be certain on the precise numbers within the “other” category. We performed free text searching for PSW to attempt to delineate this further but we prefer not to emphasize this since it is possible we are missing PSWs since it relies on free text entry.

p. 3 line 49 suggest revising to read “reported from a single centre has ranged”

Response: corrected as suggested.

p. 4 line 75. Many people will not know what a Crown corporation is. Can you use a more descriptive or generic term?

Response: Changed crown to government

p. 4 line 81 insert ‘and’ before comorbidities

Response: Added

p. 5 line 100. Is there any way or sense of how multiple occupancy addresses (apartment buildings were included and how they were handled in the data analysis?

Response: Further details on the NLP algorithm was added including the incorporation of apartment suite numbers (page 6 line 126-131): “The NLP algorithm used Python’s sklearn library. Address text was broken down into short segments with a term-frequency inverse document frequency matrix (TF-IDF). The closest proximity match within the TF-IDF matrix was returned and validated using checks for numerical portions of the address field, including suite number if available.”

p. 7 and other pages. Can you clarify if you are defining nosocomial and acquired in a hospital or acquired in the place the HCW works/from a patient. As most infections are in personal support workers, I found this confusing and the discuss about this on p.8 line 178 to the end of the paragraph took me several readings to understand and I am still not sure I am following correctly. Is all of this discussion about the fact that it is hard to be sure how HCWs were infected, or is it more complex?

Response: Based on the editors comments discussion of this variable has been substantially de-emphasized based on the degree of missing data.

6. PLOS authors have the option to publish the peer review history of their article (what does this mean?). If published, this will include your full peer review and any attached files.

Do you want your identity to be public for this peer review? For information about this choice, including consent withdrawal, please see our Privacy Policy.

Reviewer #1: Yes: Annette M. Totten

---

## [Editor Report · Decision Letter 1]

11 Dec 2020

Epidemiology, Clinical Characteristics, Household Transmission, and Lethality of Severe Acute Respiratory Syndrome Coronavirus-2 Infection among Healthcare Workers in Ontario, Canada

PONE-D-20-18932R1

Dear Dr. Schwartz,

We’re pleased to inform you that your manuscript has been judged scientifically suitable for publication and will be formally accepted for publication once it meets all outstanding technical requirements.

Kind regards,

Lamberto Manzoli, M.D., M.P.H.

Academic Editor

PLOS ONE
---

## [Editor Report · Acceptance letter]

15 Dec 2020

PONE-D-20-18932R1 

Epidemiology, Clinical Characteristics, Household Transmission, and Lethality of Severe Acute Respiratory Syndrome Coronavirus-2 Infection among Healthcare Workers in Ontario, Canada 

Dear Dr. Schwartz:

I'm pleased to inform you that your manuscript has been deemed suitable for publication in PLOS ONE. Congratulations! Your manuscript is now with our production department. 

Kind regards, 

on behalf of

Dr. Lamberto Manzoli 

Academic Editor

PLOS ONE